# ZBTB16-RARα-Positive Atypical Promyelocytic Leukemia: A Case Report

**DOI:** 10.3390/medicina58040520

**Published:** 2022-04-06

**Authors:** Laura Pardo Gambarte, Aída Franganillo Suárez, Javier Cornago Navascués, Carlos Soto de Ozaeta, Carlos Blas López, Mireia Atance Pasarisas, Rocío Nieves Salgado Sánchez, Cristina Serrano del Castillo, Raquel Mata Serna, Diego Velasco Rodríguez, José Luis López-Lorenzo, Pilar Llamas-Sillero, Laura Solán Blanco

**Affiliations:** 1Department of Hematology, Fundación Jiménez Díaz University Hospital, 28040 Madrid, Spain; laura.pardo@quironsalud.es (L.P.G.); afranganillos@gmail.com (A.F.S.); javier.cornago@hospitalreyjuancarlos.es (J.C.N.); csoto@fjd.es (C.S.d.O.); cblas@quironsalud.es (C.B.L.); mireia.atance@quironsalud.es (M.A.P.); rocio.salgado@quironsalud.es (R.N.S.S.); cserranoc@quironsalud.es (C.S.d.C.); rmatas@fjd.es (R.M.S.); diego.velascor@quironsalud.es (D.V.R.); jllopez@quironsalud.es (J.L.L.-L.); pllamas@fjd.es (P.L.-S.); 2Health Research Institute-Fundación Jiménez Díaz University Hospital, 28040 Madrid, Spain

**Keywords:** acute promyelocytic leukemia, promyelocytic leukemia zinc finger protein, retinoic acid receptor alpha, all trans retinoic acid, case report

## Abstract

Background: The majority of patients with acute promyelocytic leukemia (APL) manifest a specific chromosomal translocation t(15;17)(q22;q21), characterized by the fusion of RARA and PML genes. However, a proportion of APL cases are due to variant translocations, being t(11;17) (q23;q21) the most common amongst them. With the major exception of ZBTB16-RARA t(11;17) APL, these variant APL cases present similar morphological features as classic APL and are characterized by a lack of differentiation response to retinoids. Case summary: We describe the case of variant APL with the ZBTB16-RARA fusion gene, showing a distinct morphology of classical APL, characterized by crystalline intracytoplasmic inclusions in both peripheral blood (PB) and bone marrow (BM) patients’ blasts. Our patient was treated with two courses of intensive chemotherapy, initiating maintenance treatment with all-trans retinoic acid (ATRA) on day twenty-eight of the second course. Our patient achieved complete remission (CR) once the intensive chemotherapy was combined with ATRA.Conclusions: This is the second case described of APL with t(11;17) that showed crystalline intracytoplasmic inclusions. The finding of these morphological features may suggest the presence of a variant translocation with RARA, being that both cases described are related to the presence of t(11;17). Despite induction treatment with intensive chemotherapy that included a seven-day continuous treatment with cytarabine (200 mg/m^2^), plus daily idarubicin (12 mg/m^2^) during the first three days, our patient did not achieve complete remission (CR) until scheduled 3 + 7 regimen combined with ATRA treatment was established. This observation suggests that ATRA may be partially effective in some ZBTB16-RARA APLs.

## 1. Introduction

APL is a distinct subtype of AML, typified by the presence of the balanced translocation t(15;17)(q22;q21), which leads to the formation of the PML-RARA fusion gene [1]. APL is characterized by the presence of atypical promyelocytes in the BM and favorable overall survival with retinoid-based therapy [1]. A small percentage of APL cases lack the classic t(15;17) and show variant translocations involving RARA, being t(11;17)(q23;q21) the most widely identified [2]. This variant translocation generates the ZBTB16-RARA fusion gene and is associated with resistance to ATRA [3]. APL cases with t(11;17) are characterized by an atypical morphology, showing blasts with regular nuclei, an absence of Auer rods, and an increased number of pseudo-Pelger–Huët neutrophils [4].

## 2. Case Report

We present the case of a 62-year-old male patient with a personal history of gout, who was referred to our hospital because of pancytopenia during a routine blood test. The patient was asymptomatic, and there were no findings on physical examination. Full blood count revealed leukocytes of 2840/mm^3^ (neutrophils 700/mm^3^), hemoglobin of 11.3 g/dL and platelets of 73,000/mm^3^. Coagulation screening was normal, with a prothrombin time of 13.4 s, activated partial thromboplastin time of 27.2 s, and levels of derived fibrinogen of 210 mg/dL. Renal and hepatic function tests were normal.

PB smear confirmed thrombocytopenia and revealed the presence of anisopoikilocytosis with a count of two erythroblasts per 100 leukocytes. Hypergranular myelocytes were observed, accounting for 8% of total cell count, as well as pseudo-Pelger–Huët neutrophils. Multiparameter flow cytometry (MFC) of a PB sample was performed, detecting 26% of autofluorescent promyelocytes that expressed MPO+, CD33++, CD13+, CD117± and lacking CD56 and CD34. 

On BM aspirate, the majority of cells were abnormal promyelocytes with round, non-lobulated nuclei without nucleoli and hypergranular cytoplasm. Most of these cells lacked Auer rods (Figure 1a), and some promyelocytes showed intracytoplasmic inclusions similar to those previously described by Dowse et al. [5] (Figure 1b). Dyserythropoiesis and granulocytic dysplasia were also described (Figure 1c).

MFC of the BM showed a predominance of premature granulocytic cells, detecting 59% of atypical promyelocytes with positivity for CD33++, CD13+, MPO+, CD64±, CD117±, and negativity for CD15, HLA-DR, CD11b, CD56, CD203c, and CD34.

An early fluorescence in situ hybridization (FISH) study with *Vysis LSI PML/RARA dual-colour dual-fusion translocation probe kit (Abbott Molecular, Les Plaines, IL)*, showed three green FISH signals suggesting a variant RARA translocation. RARA rearrangement was confirmed using a RARA break-apart FISH probe *(NIMFISH)* (Figure 2a).Cytogenetic analysis was performed on 24- and 48-h in vitro cultures of BM cells. GTG banding techniques were applied, obtaining 10 metaphases, which all showed a male karyotype 46,XY,t(11;17)(q23;q21)[10] (Figure 2b). These findings suggested the presence of a *ZBTB16/RARA* fusion gene.

A standard RT-PCR approach did not detect the presence of bcr-1, bcr-2, and bcr-3 transcripts corresponding to *PML/RARA* rearrangement. Due to our cytogenetic findings, we developed a real-time quantitative PCR, using the reverse primer located in RARA exon 3 in conjunction with the primer located within *ZBTB16* (exon 3 and 4), similar tothose described by Jovanovic et al. [6]. RT-qPCR reaction was run on with the expression of normalized leukemic fusion transcripts of the *ABL* control gene using the ΔCt method. In our case, the chromosome 11 breakpoint fell within ZBTB16 intron 3, leading to the retention of two zinc fingers (2ZF) in the ZBTB16 moiety of the ZBTB16-RARA fusion protein. Furthermore, an NGS panel was set up, detecting additional mutations in *FLT3, IDH2* and *SRSF2*.

With a diagnosis of atypical APL, the patient was started on intensive chemotherapy following a PETHEMA pre-established regimen for AML that included a seven-day continuous treatment with cytarabine (200 mg/m^2^), plus daily idarubicin (12 mg/m^2^) during the first three days (3 + 7 regimen). As per protocol, on day thirty of induction therapy, a BM aspirate was repeated, which reported morphological evidence of residual disease showing the presence of 7% of atypical promyelocytes. Cytogenetic testing found persistence of t(11;17) in 10% of all analyzed metaphases, and ZBTB16-RARA rearrangement was detected at 11% by RT-PCR.

Following these results, reinduction therapy with a new 3 + 7 course was scheduled. We decided to initiate maintenance treatment with ATRA on day twenty-eight of this reinduction treatment. BM aspirate was repeated on day fourty-eight, showing the first complete morphological remission, with negative minimal residual disease (MRD) by multiparameter flow cytometry. These results were consistent with cytogenetic tests showing a normal karyotype and with molecular studies that did not find the *ZBTB16-RARA* rearrangement.

First consolidation treatment was given with high-dose cytarabine (HiDAC), consisting of six doses (1 g/m^2^/12 h) on days 1, 3, and 5. During this time, the patient continued ATRA therapy daily. BM aspirate was repeated on day +46, persisting with MRD and not finding the *ZBTB16-RARA* rearrangement. The patient then received two more cycles of HiDAC with a new bone BM showing the absence of disease after each cycle. Maintenance therapy with ATRA was reduced to fifteen days per month after the second HiDAC consolidation.

At the date of publication, the patient continues in CR and continues receiving ATRA as maintenance therapy fifteen days per month until further evaluation.

This case has been recently included in the PETHEMA REGISTRY [7].

## 3. Discussion

APL cases with t(11;17) are characterized by an atypical APL morphology, showing a predominance of blasts with regular nuclei, hypergranulation of most cells, absence of Auer rods, and an increased number of pseudo-Pelger–Huët neutrophils [2]. Characteristically, our patient’s blast cells in both, PB and BM smears, showed crystalline intracytoplasmic inclusions akin to those described by Dowse et al. [5]. Despite no other reports describing these inclusions, the finding of these morphological features may suggest the presence of a variant translocation with RARA, being that both cases described are related to the presence of t(11;17).

Our patient had an abnormal karyotype with t(11;17)(q23;q21), and molecular studies showed the presence of the *ZBTB16-RARA* fusion gene. In the literature, there have been described nineteen patients with a karyotype with two or more cytogenetic alterations, which can be related to a poorer prognosis [7].

An NGS approach detected mutations in *FLT3, IDH2* and *SRSF2.* Both FLT3 and SRSF2 have been described in the literature related to t(11;17) APLs. IDH2 has not been previously described in relation to this translocation [8]. Another published report describes the finding of a mutated gene *CEBPA* [2]. A previous study demonstrated that the RARA-ZBTB16 gene inhibits myeloid cells differentiation through its interaction with *CEBPA* [9]. However, the possible prognostic value of our findings cannot yet be established as the cases of t(11;17) APLs with the mutation profile described are limited.

From a therapeutic point of view, t(11;17) APL is characterized by poor response to ATRA, resulting in an unfavorable prognosis [1,4]. It was at first suggested that this resistance to treatment resulted from the ability of the ZBTB16 moiety to bind corepressors, but it has been proven that ZBTB16-RARA leukemia cells can fully differentiate on ATRA treatment [10]. A better CR has been observed in the patients who received ATRA plus intensive chemotherapy [7]. Due to these findings, our patient received induction treatment with an intensive chemotherapy regimen following standard AML protocols. Despite blast decreases observed in the BM aspirate by day 30, our patient did not achieve CR until the scheduled 3 + 7 regimen combined with ATRA treatment was established. This observation suggests that ATRA may be partially effective in some PLFZ-RARA APLs. The duration of ATRA administration in patients with ZBTB16-RARA APL remains uncertain. In our case, ATRA treatment was added as maintenance after reinduction, though there is still a lack of evidence to support this decision. In our case, ATO was not considered as further treatment due to the achievement of complete remission after the reinduction regimen.

As far as we know, only four patients underwent allogeneic stem cell transplantation (HSCT). All four had been in CR for over one year following transplantation when reported [11,12], so it seems reasonable to consider allogeneic HSCT in first CR if the patient has a suitable donor, contrary to what is suggested in classic APL relapse, where protocols recommend autologous HSCT. One reported patient with ZBTB16-RARA APL underwent an autologous HSCT but died twenty-three months later in a second relapse [12].

The experience of MRD detection in ZBTB16-RARA cases is limited, but it seems analogous to PML-RARA tracking in patients with the classic variant of the disease. The quantitative RT-PCR approach allows monitorization and confirms molecular response to treatment [6]. Our patient has achieved their first CR with negative MRD as observed through qRT-PCR and is now being followed by our Hematology team as an outpatient. MRD can be accurately followed through molecular studies, so HSCT has been delayed for the time being, though it remains a possibility as consolidation treatment after a second CR if the patient relapses.

Overall, early recognition of APL cases with t(11;17) is critical in patient survival due to ATRA resistance and poorer prognosis. A combination of morphology analysis, clinical features, and genetic tests is necessary to achieve rapid identification.

## 4. Conclusions

Variant APL cases present similar morphological features as classic APL, with the exception of ZTB16-RARA t(11;17). This is the second case of APL with t(11;17) that showed this characterized morphology based on crystalline intracytoplasmic inclusions in PB and BM patient’s blasts. Due to the limited cases described in the literature, the value of these findings can not be established, but we think that the finding of these morphological features may suggest the presence of a variant translocation with RARA, being that both cases described are related to the presence of t(11;17).

Our patient did not achieve complete remission (CR) until ATRA was added to the chemotherapy regimen. This observation suggests that ATRA may be partially effective in some ZTB16-RARA APLs.

## Figures and Tables

**Figure 1 medicina-58-00520-f001:**
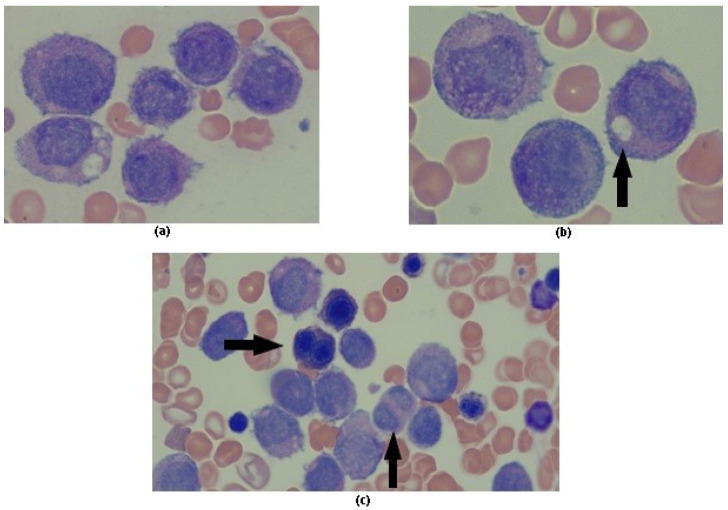
Bone marrow sample. (**a**) Hypergranular leukemic promyelocytes; (**b**) Intracytoplasmic inclusions; (**c**) Dyserythropoiesis and granulocytic dysplasia.

**Figure 2 medicina-58-00520-f002:**
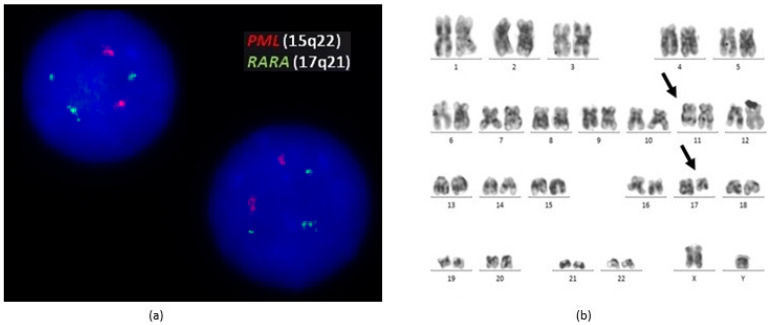
(**a**) FISH analysis of the patient. The Vysis probe set showed two signals on chromosome 15 (PML) and three signals on chromosome 17 (RARA), suggesting RARA rearrangement; (**b**) Karyotype analysis of the bone marrow (GTG bands): 46,XY,t(11;17)(q23;q21)[10].

## Data Availability

No new data were created or analyzed in this study. Data sharing is not applicable to this article.

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
