# Peer review of "ZBTB16-RARα-Positive Atypical Promyelocytic Leukemia: A Case Report"

_medicina, 2022, doi:10.3390/medicina58040520_

Round 1
Reviewer 1 Report
- Because of the limited APL cases with t(11;17) that showed crystalline intracytoplasmic inclusions, the conclusion that "Our findings suggest that t(11;17) cases have morphological differences with other APL cases which may aid in its differential diagnosis." was insufficiency.
- PLZF and ZBTB16 need to be unified in the manuscript.
- The detection of PLZF-RARA should be demonstrated by PCR and Sanger-sequence, and the corresponding results should be presented.
- Some PLZFs were written as PLFZ. "PML" in line 73 was written as "PLM".
- The presentation of karyotype result in line 108 was not standard.
Author Response
Response to Reviewer 1 Comments
Point 1: Because of the limited APL cases with t(11;17) that showed crystalline intracytoplasmic inclusions, the conclusion that "Our findings suggest that t(11;17) cases have morphological differences with other APL cases which may aid in its differential diagnosis." was insufficiency.
Response 1: We appreciate this comment and agree that due to the limited cases described in the literature we can not conclude that all the APL cases with t(11;17) will have the crystalline intracytoplasmic inclusions that we observed in our case. Based in yours suggestions, we have change de way to express this idea in the manuscript as you can see in the abstract (line 28-29) and in the conclusions section (line 174-175)
Point 2: PLZF and ZBTB16 need to be unified in the manuscript.
Response 2: Following your suggestion, we have unified PLZF and ZBTB16 in the manuscript.
Point 3: The detection of PLZF-RARA should be demonstrated by PCR and Sanger-sequence, and the corresponding results should be presented.
Response 3: We really aprecciate your comment. As it has been described in the main manuscript, we first did an eary FISH that showed a variant RARA translocation. We did then a review of the literature and we developed a RT-PCR using one of the primers described by Jovanovic et al. As we confirmed the presence of ZBTB16-RARA fusion transcript by RT-PCR we did not do a Sanger-sequence.
Point 4: Some PLZFs were written as PLFZ. "PML" in line 73 was written as "PLM".
Response 4: Following your suggestion, we have reviewed the spelling of the genes and we have done the aproppiate changes.
Point 5: The presentation of karyotype result in line 108 was not standard.
Response 5: According with your suggestion, we have changed the karyotipe presentation of this line as you can see in line 108.

Reviewer 2 Report
Very good written, interesting case. The diagnosis and the coltrolls were made with accepted molecular methods.
Author Response
Response to Reviewer 2 Comments
We appreciate the interest shown in reviewing this work. We have carefully reviewed English languaje.
We have done some changes in the manuscript in order to improve the quality of the case report. You can see some improvements that have been done on the line 106 or in the 128.
Reviewer 3 Report
Acute promyelocytic leukemia (APL) with the variant t(11; 17) cytogenetic alteration represent a very rare APL subtype, with poorer prognosis compared to the classic bearing the t(15;17).
On this paper the authors provided a precise description of a case with this rare translocation variant.
The paper is well done, although confirmatory in many parts, and there are many points to clarify.
- The authors underline the importance of morphological differences compared to classical APL, and report high flow cytometric autofluorescence in this subtype. This fact, along with the crystalline cytoplasmic inclusions may have “diagnostic” weight in this leukemic subtype?
- Blasts do not express CD56, always described in this leukemic variant. Can the authors suggest an explanation?
- The NGS evaluation detected FLT3, IDH2, and SFSR2mutations. The literature data reported higher frequency of TET2, CSF3R mutations compared to t(15;17) APL, and no FLT3 mutations. How did the authors explain this fact? Moreover high incidence of mutations of ARID1A, belonging to chromatin remodeling complexes was previously described. Did the authors evaluate them in the NGS panel?
- The reported only partial response after the first 3+7 induction therapy. Morphologic and MRD response was obtained only after ATRA maintenance. Pleas add a more detailed comment on this fact.
Author Response
Response to Reviewer 3 Comments
Point 1: The authors underline the importance of morphological differences compared to classical APL, and report high flow cytometric autofluorescence in this subtype. This fact, along with the crystalline cytoplasmic inclusions may have “diagnostic” weight in this leukemic subtype?
Response 1: We appreciate the interest shown in our manuscript. Due to the limited cases described in the literature, we can not conclude that all the APL cases with t(11;17) will have the crystalline intracytoplasmic inclusions that were observed in our case. We think that the finding of these morphological features may suggest the presence of a variant translocation with RARA, being the both cases described in the literature, related with the presence of t(11;17). The presence of these variant translocation should be confirmed with RT-PCR. We have changed the way to express it in the abstract (line 28-29) and in the conclusions section (line 174-175).
Point 2: Blasts do not express CD56, always described in this leukemic variant. Can the authors suggest an explanation?
Response 2: We really appreciate your comment. As we have mentioned before, due to the infrecuency of this type of leukemia it is unlikely to established a inmunophenotype characterization in this entity. In our case, the inmunophenotype was negative for CD56 both in blood and in the bone marrow, contrary for what it is drescibed in the literatura until now. We did not find any reason to the justify our findings.
Point 3: The NGS evaluation detected FLT3, IDH2, and SFSR2 mutations. The literature data reported higher frequency of TET2, CSF3R mutations compared to t(15;17) APL, and no FLT3 mutations. How did the authors explain this fact? Moreover high incidence of mutations of ARID1A, belonging to chromatin remodeling complexes was previously described. Did the authors evaluate them in the NGS panel?
Response 3: Thank you for you appreciation. Reviewing the literature, we found a manuscript that describes the mutation profile of 7 cases of ZBTB16-RARA APLs. In this review, there was one case described with FLT3 mutation and another one with SRSF2. There were no cases of IDH2 mutated. In our panel, we could not evaluated the presence of ARID1A.
Point 4: The reported only partial response after the first 3+7 induction therapy. Morphologic and MRD response was obtained only after ATRA maintenance. Pleas add a more detailed comment on this fact.
Response 4: We really appreciate you comment. After the first 3+7 induction therapy we did a BM aspirate where we found that the patient had a partial response with persistance of 7% of atypical promyelocites. After this response, we decided to continue with a second 3+7 reeinduction therapy, but we iniciate maintenante treatment with ATRA on day 28. We did a new BM aspirate tweenty days after, where we observe a complete response (with MRD negative by multiparameter flow cytometry and lack of ZBTB16-RARA rearrangement). We have changed the way we express it in the lines 106-108.
